# The African Swine Fever Epidemic in Wild Boar (*Sus scrofa*) in Lithuania (2014–2018)

**DOI:** 10.3390/vetsci7010015

**Published:** 2020-01-30

**Authors:** Petras Mačiulskis, Marius Masiulis, Gediminas Pridotkas, Jūratė Buitkuvienė, Vaclovas Jurgelevičius, Ingrida Jacevičienė, Rūta Zagrabskaitė, Laura Zani, Simona Pilevičienė

**Affiliations:** 1Department of Veterinary Pathobiology, Lithuanian University of Health Sciences, LT-44307 Kaunas, Lithuania; 2Management Department, National Food and Veterinary Risk Assessment Institute, 10 LT-08409 Vilnius, Lithuania; gediminas.pridotkas@nmvrvi.lt; 3Emergency Response Division, State Food and Veterinary Service, LT-07170 Vilnius, Lithuania; 4Serology Department, National Food and Veterinary Risk Assessment Institute, 10 LT-08409 Vilnius, Lithuania; jurate.buitkuviene@nmvrvi.lt (J.B.); ruta.zagrabskaite@nmvrvi.lt (R.Z.); 5Molecular Biology and GMO Department, National Food and Veterinary Risk Assessment Institute, 10 LT-08409 Vilnius, Lithuania; vaclovas.jurgelevicius@nmvrvi.lt (V.J.); simona.pileviciene@nmvrvi.lt (S.P.); 6Biology Department, Vytautas Magnus University, LT-44248 Kaunas, Lithuania; 7Virology Department, National Food and Veterinary Risk Assessment Institute, 10 LT-08409 Vilnius, Lithuania; ingrida.jaceviciene@nmvrvi.lt; 8International Animal Health Team, Friedrich-Loeffler-Institut, 17493 Greifswald–Insel Riems, Germany; Laura.Zani@fli.de

**Keywords:** wild boar, African swine fever, prevalence, spread

## Abstract

In January 2014 the first case of African swine fever (ASF) in wild boar of the Baltic States was reported from Lithuania. It has been the first occurrence of the disease in Eastern EU member states. Since then, the disease spread further affecting not only the Baltic States and Poland but also south-eastern Europe, the Czech Republic and Belgium. The spreading pattern of ASF with its long-distance spread of several hundreds of kilometers on the one hand and the endemic situation in wild boar on the other is far from being understood. By analyzing data of ASF cases in wild boar along with implemented control measures in Lithuania from 2014–2018 this study aims to contribute to a better understanding of the disease. In brief, despite huge efforts to eradicate ASF, the disease is now endemic in the Lithuanian wild boar population. About 86% of Lithuanian’s territory is affected and over 3225 ASF cases in wild boar have been notified since 2014. The ASF epidemic led to a considerable decline in wild boar hunting bags. Intensified hunting might have reduced the wild boar population but this effect cannot be differentiated from the population decline caused by the disease itself. However, for ASF detection sampling of wild boar found dead supported by financial incentives turned out to be one of the most effective tools.

## 1. Introduction

The current African swine fever (ASF) epidemic in Europe and Asia is caused by a highly virulent strain belonging to genotype II. ASF was introduced into Georgia in 2007 from where the disease spread throughout the Caucasus and the Russian Federation (RF), reaching Ukraine in 2012 and Belarus in 2013. In January 2014, ASF arrived in the eastern part of the European Union (EU) when infected wild boars were detected in Lithuania. Later during the same year, the disease was reported from Poland, Latvia, and Estonia. In Lithuania as well as in the other Baltic States and Poland ASF became endemic in the wild boar population whereas outbreaks occurring in domestic pigs could be contained preventing extensive secondary spread [1]. 

Retrospectively, two main patterns of disease spread within Europe became evident: (i) slow local spread by direct contact between wild boar and (ii) huge jumps of more than hundreds of kilometers (e.g., virus spread to the Czech Republic and Belgium). For local spread by direct contact a speed of 2–5 km/month has been calculated [2]. Humans were recognized as the main cause of both long-distance transmission and virus introduction into uninfected wild boar habitats and domestic pig farms [3].

When ASF entered Lithuania in 2014 two main epidemiological scenarios were forecasted: either the disease would spontaneously fade out from the local wild boar population due to the high case fatality rate or, alternatively a rapid epidemic wave like experienced with rabies would start moving westward very rapidly, affecting large areas of Europe [4]. However, both hypotheses proved to be wrong. The disease did not fade out nor did it show a rabies-like fulminant epidemic wave behavior. On the contrary, the infection survived locally in wild boar populations with a steady, low prevalence (Table 1) [5].

This article aims to review in a descriptive way the course of ASF in wild boar populations in Lithuania along with the implemented control measures. 

## 2. Materials and Methods 

A retrospective prevalence analysis was conducted regarding spatial and temporal distribution of ASF in wild boar in Lithuania during 2014–2018. Data were obtained from National Databases for Animal Disease Notification of the Emergency Response Department at the State Food and Veterinary Service of the Republic of Lithuania. Sample record contains sampling date and time, place, sample name, code, testing methods, animal species, sex, age, name of territorial competent authority and the owner of the sample, type of surveillance, sample acceptance, and testing date and test result. ASF prevalence was analyzed in terms of animals found dead or hunted during the different years (time), seasons (winter, spring, summer, and autumn) and regions (regional municipalities, elderships). Sex and age (<12 months, 12–24 months, >24 months, and unidentified age) of wild boar found dead and hunted were also included into the analyses. 

Lithuania is divided into three levels of administrative divisions. The first-level division consists of 10 counties. These are sub-divided into 60 municipalities, which in turn are further sub-divided into over 546 smaller groups, known as elderships. The largest municipality by land area has 2216 km^2^, while the smallest has 40 km^2^. An eldership may comprise a very small region consisting of few villages, one single town, or a part of a big city. Elderships vary in size and population depending on their location and nature. The largest eldership by land area is 550 km^2^, while the smallest is 1.47 km^2^.

A wild boar was considered ASF infected either after a positive polymerase chain reaction (PCR) result or after a positive antibody test (enzyme-linked immunosorbent assay (ELISA) and/or immunoperoxidase test (IPT)). All laboratory tests were performed at the Lithuanian National Food and Veterinary Risk Assessment Institute following international OIE standard operating procedure.

For laboratory testing tissue samples were collected from animals found dead or killed in road incidents in the entire country. From hunted animals in the infected regions (listed in Parts II and III of the Commission Implementing Decision 2014/709/EU) blood samples were collected. From animals hunted outside the infected regions (listed in Part I of the Commission Implementing Decision 2014/709/EU) or in non-infected areas, either blood or tissue samples were taken. The different restriction zones are mapped according the Commission Implementing Decision 2014/709/EU. Sample size in non-infected areas was calculated according EU legislation (minimum 56 samples per regional municipality) [6].

Biosecurity measures were applied during the sampling and safe disposal of the carcasses to prevent the further spread of virus. Every wild boar found dead, killed in the road incident and/or hunted in the infected area had to be sampled and tested. Hunters were trained to apply biosecurity measures during sampling and safe disposal of wild boar animal by-products. Hunted wild boar carcass remains in the hunting ground retained locked in the refrigerator until a negative test result received. All hunted wild boars with a positive test and wild boar found dead were safely disposed by burring in the land within the hunting ground area or putting into the specially designated animal waste pit for safe disposal of animal by-products under official control. People searching wild boar carcasses were obliged to notify to the competent authority on finding of wild boar carcass and not touch it.

ASF spreading rate was calculated for each year. In 2014, it was calculated how far the disease had spread from its first detection site. In the following years, it was calculated how far the disease had spread from the last place it was detected. The distance of the disease spread was measured in kilometers and in different directions from the randomly selected five points where the disease was detected in the last year to the sites where the disease was detected in the current year. The spread rate was calculated by dividing the average of five measurements by 12 months.

Descriptive data analysis was performed using Microsoft Excel 2007 (Microsoft) and SigmaPlot for Windows version 11.0 (Systat Software, Inc., San Jose, California).

## 3. Results

### 3.1. Wild Boar Hunting Bag in Lithuania (2013–2019)

Generally, the number of hunted wild boar fluctuates over the years but there is a considerable and steady decrease from 2014 to 2019. Exceptionally, during the hunting season of 2017 a slight increase of the wild boar hunting has been seen (Figure 1).

### 3.2. Proportion of ASF Positive Wild Boar in Lithuania (2014–2018)

The number of positive wild boar samples increased over the years. In 2017, the overall amount of samples of found dead wild boar was more than three times higher than in 2016. At the same time the number of ASF positive wild boar found dead increased more than 5 times. The proportion of PCR positive samples was up to hundred times higher in samples of wild boar found dead compared to samples of hunted wild boar (Figure 2 and Appendix A). A considerable increase of antibody positive samples in hunted wild boar was seen in 2017 and 2018 (Appendix A). 

### 3.3. ASF Distribution by Sex and Age Class of Wild Boar in Lithuania (2014–2018)

In the group of wild boar found dead slightly more females were tested ASF positive than males. In around 70 per cent of the carcasses of wild boar found dead that has been tested positive for ASF the sex could not be determined. In the group of hunted wild boar slightly more males were found positive (Figure 3). Concerning age distribution, most positive animals were detected in the age group of 12–24 months in both, hunted and found dead wild boar. In found dead wild boar the age group could not be determined in almost half of the carcasses that have been tested positive for ASF (Figure 4).

### 3.4. Seasonal Distribution of ASF in Lithuania during 2014–2018

The overall proportion of PCR positive wild boar found dead increased from 2014 to 2018. In 2017 and 2018 the proportion of positive samples was slightly higher in winter season. The amount of PCR positive samples was slightly lower in autumn in all years except 2017 (Figure 5). The proportion of antibody positive hunted wild boar increased in 2017 and 2018 with a peak (2.59%) in autumn 2018 (Appendix A and Appendix A). 

### 3.5. Spatial Distribution and Spreading Speed of ASF in Lithuania (2014–2018)

The ASF epidemic in Lithuania started 2014 in the eastern part of the country and spread westwards. Apart from few municipalities in the very western part of Lithuania ASF cases occurred in wild boar all over the country. From 2014 to 2018 ASF was reported in 42 out of 60 regional municipalities and in 300 out of 549 elderships (Figure 6 and Appendix A). A considerable increase in the number of affected municipalities and elderships was seen in 2017 and 2018. On municipality level about 25% of the Lithuanian territory was affected in 2014 and 84% in 2018. On eldership level, an increase from 7% in 2014 to 53% in 2018 could be seen (Table 1). More than one third of all ASF cases occurred in Anykščiai, Panevėžys, and Ukmergė municipalities (Appendix A). 

The average spreading rate to non-affected areas, in 2014 was approx. 5 km/month, in 2015—approx. 3 km/month, and in 2016—approx. 2 km/month. In 2017 and 2018 the average spread rate was higher: 7 km/month (2017) and 8 km/month (2018). The total average of spreading rate over 5 years has been calculated to be 5 km/month.

### 3.6. Measures Undertaken in Lithuania to Control ASF

Since the beginning of the epidemic in 2014, ASF affected areas have been regionalized following EU legislation (Commission implementing Decision (2014/709/EU) and sampling and surveillance strategies have been adapted accordingly. 

In addition to measures provided by EU directive 2002/60/EC, national disease control measures have been implemented. From November 2015 incentives for hunting wild boar were implemented to reduce the wild boar population by intensified targeted hunting of adult and sub-adult females. Hunters received 50 € for a 12–24 months old female and 100 € for a female over 24 months of age. From October to December 2017 the incentives were raised up to 300 € for wild boar females older than 24 months. In addition, wild boar hunting was allowed all-the-year. At the beginning of 2016 the Lithuanian government started to pay 30 € to people reporting dead wild boar. Consequently, people actively searched for wild boar carcasses and the number of reports increased considerably. A further financial incentive for finding and reporting of dead wild boar on ASF affected hunting grounds was introduced in September 2017. Hunters were paid additionally for carcass disposal by burying (Figure 7).

## 4. Discussion

To understand the ASF situation in Lithuania during 2014–2018, it is important to consider all administrative measures taken by the competent authorities to monitor, control and eradicate the disease in wild boar populations [7,8]. By only judging laboratory results and without taking into account the implemented control measures and sampling strategies, a misleading picture might be obtained. The compensation system established by the Lithuanian Government can be seen as a major driver to obtain samples for effective surveillance. In particular sampling of dead wild boar has been shown to be crucial for early disease detection [1,9]. As sampling carcasses is cumbersome and unpleasant, incentives can help to raise the willingness of people towards it. In case of Lithuania, the paying of incentives (implemented in 2016) lead to a considerable increase in positive samples of dead wild boar. 

The proportion of positive animals, found dead and hunted, increased from year to year and nearly doubled from 26% in 2015 to 51% in 2016 (Figure 2). However, the prevalence of positive animals in the group of hunted animals remained below 2% while the ASF prevalence in wild boar found dead ranged between 62% and 83%. The high proportion of positive cases in wild boar found dead in combination with the very low proportion of seropositive animals (<2%) are indicative for a high case fatality rate as described for the ASF strain circulating in Europe [10]. This data is in line with recent findings in other affected countries [1,11,12]. These findings clearly emphasize the important role of passive surveillance by sampling and testing of wild boar found dead. Passive surveillance proved to be the most effective tool for early ASF detection and for defining areas affected by the disease [1,2]. The relatively low prevalence (<2%) of PCR positive animals in hunted wild boar would not favor early detection of ASF neither contributing to define precisely the infected areas. 

The decrease in wild boar hunting bags in Lithuania over the years can be explained by different factors. If the hunting bag is taken as a proxy for the Lithuanian wild boar population it would indicate a considerable decrease in the population [13]. With the high case fatality of ASF spreading throughout the country a decrease in the population due to the ASF epidemic would be more than expected. The slight increase of the wild boar hunting bag in 2017 can be traced back to the incentive program for intensified hunting of female wild boar that has been implemented during this hunting season. The overall hunting bag was more or less balanced between males and females. According to EU ASF control strategy 50% of the hunted animals should be female [14]. Therefore, it can be considered that the applied compensation system to reduce female wild boar did reach some expectations in sense of an increased hunting activity. However, the reducing effect of intensified hunting on the wild boar population could not be quantified and distinguished from the reducing effect of the fatal ASF epidemic. The population reducing effect of the year-round wild boar hunt authorized in 2014 could not be measured for the same reason. Generally, the effect of recreational hunting on wild boar population density seems to be limited [15]. 

During the first three years of the epidemic (2014–2016) only a relatively moderate spreading of ASF was observed in eastern and central parts of Lithuania. A considerable spatial spread towards East–South, East–North, and central parts of the country occurred in 2017 and 2018. So far, factors leading to this sudden expansive spread remain unclear. One factor influencing the willingness of reporting wild boar carcasses and thereby the number of detected positive wild boar could be the incentives implemented 2017 (see Figure 7). The three mainly affected municipalities (Anykščiai, Panevėžys, and Ukmergė) had in common a dense wild boar population on commercial hunting grounds. Consequently, there was a high number of susceptible animals along with many people capable and willing to search for carcasses.

In literature, it is reported that ASF spreads within European wild boar populations with an average speed of 2–5 km/month [2]. For Lithuania, an average spreading rate of 5 km/month has been calculated. However, it is impossible to differentiate how much this spread was driven by humans or by direct contact of wild boar. Around 56% of all ASF cases occurred in eight municipalities. Wild boar densities, habitat characteristics, or hunting activities influenced spatial and temporal distribution of ASF spread. Detailed analysis of all ASF cases showed that many elderships located inside infected municipalities seem to be not affected by ASF. Taking the sizes of infected elderships, then ASF affected only 6.7% of the Lithuanian territory in 2014, 13.7% in 2015, 16.5% in 2016, 32.8% in 2017, and 53.4% in 2018. This suggests that some wild boar subpopulations remained uninfected for several months or years while wild boar populations in neighboring areas were affected by ASF. These findings are indicative for a rather low contagiosity and a relatively slow spread of ASF. Nonetheless, it cannot be excluded that ASF positive wild boar were present in those areas but not detected despite intensified surveillance measures. On the other hand, once affected municipalities remain positive throughout the whole monitored period. This reflects the endemic character of ASF in the Baltic States [2]. If this trend will continue over the next years is uncertain. Data from Estonia suggests a decline in the incidence of ASF [16].

The seasonal analysis of ASF positive wild boar cases does not show any constant pattern. In 2016 a peak of positive animals is seen in summer and winter season. This reflects the seasonality reported for ASF positive wild boar [1] but has not seen any more in other years included in the analysis. In 2017 and 2018 most ASF positive wild boar found dead have been detected in winter season. This could be explained by the traditionally increased hunting activity during winter seasons and the thereby enhanced chance for carcass detection. In addition, the implemented incentives for sampled dead wild boar could have contributed. In addition, the decomposition time of carcasses in winter is much longer than in summer and carcasses are much longer present in the environment [17]. 

Regarding sex distribution of ASF positive animals slightly more males were tested positive in the group of hunted animals and more females in the group of animals found dead. Furthermore, for 70% of the carcasses no sex could be identified most likely due to advanced decomposition state. However, this data does not indicate that one sex group is more susceptible for ASF than the other. 

The age group of animals 12–24 months old had the highest proportion of ASF positive animals (35% in the group of carcasses compared to 26% and 28% of other age groups). These findings can be interpreted in different ways. This particular age group could be more susceptible for ASF infection which is not in line with literature [18]. On the other hand, this particular age group might be more under risk to get infected with ASF due to larger roaming activities [19]. It should also be considered that, according to literature, yearlings represent the highest proportion in hunted wild boar as they are more attractive for hunters compared to piglets that represent the largest age-class in the overall wild boar population [20]. Hence, the lower proportion of ASF positive young wild boar (<12 months) might be explained by the fact that this age group is underrepresented in the hunting bag and has a generally smaller survival rate as piglets are further exposed to risks like starvation, hypothermia and predation [21]. However, in literature no age preference for ASF infection is mentioned [18]. 

## 5. Conclusions

To sum up the ASF epidemic in Lithuanian wild boar during 2014–2016 it can be concluded that during the first three years ASF was spreading moderately affecting less than 50% of municipality areas whereas, a faster spread of ASF was noticed in 2017 and 2018 affecting up to 84% of the Lithuanian territory. Despite intensified hunting efforts, the expansive disease spread led to a considerable decline in the wild boar hunting bag indicating a reduced wild boar population. Furthermore, passive surveillance accompanied by financial incentives for the reporting of wild boar carcasses has been shown to be crucial for ASF detection in wild boar.

## Figures and Tables

**Figure 1 vetsci-07-00015-f001:**
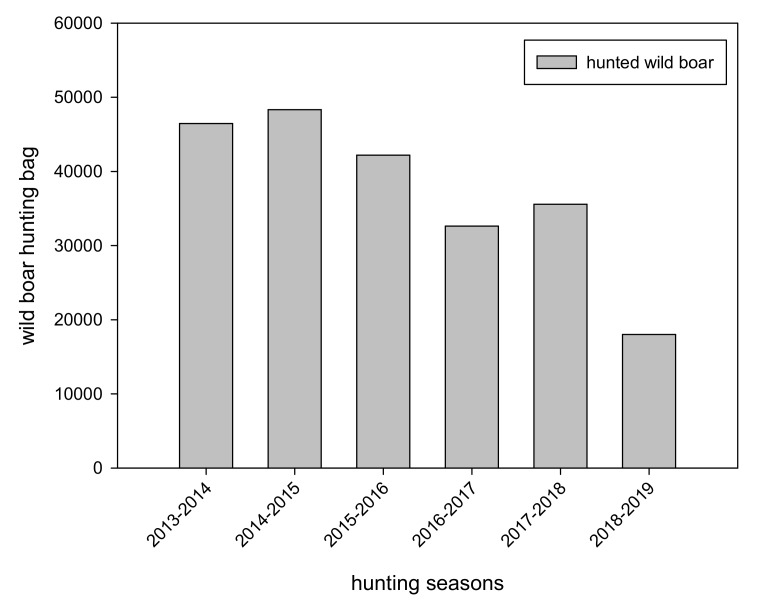
Hunting bag data of wild boar hunted in Lithuania (each hunting season is from April 15 to April 14 of the following year).

**Figure 2 vetsci-07-00015-f002:**
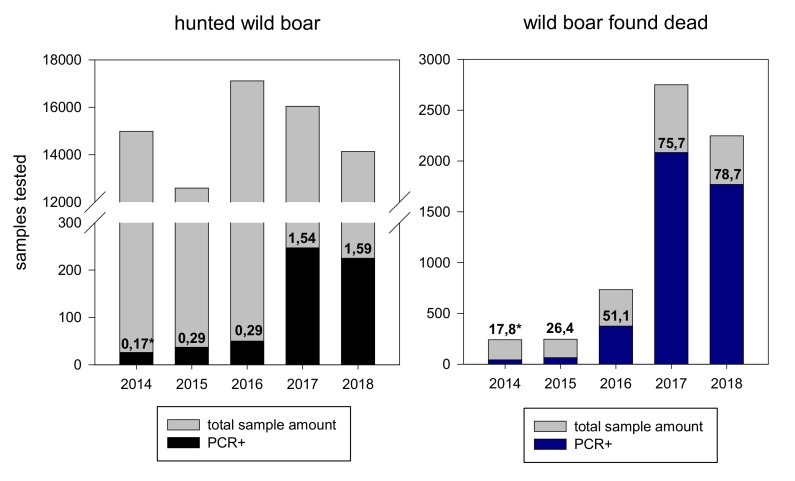
Samples of hunted wild boar (**left**) and wild boar found dead (**right**). * proportion of PCR positive samples [%].

**Figure 3 vetsci-07-00015-f003:**
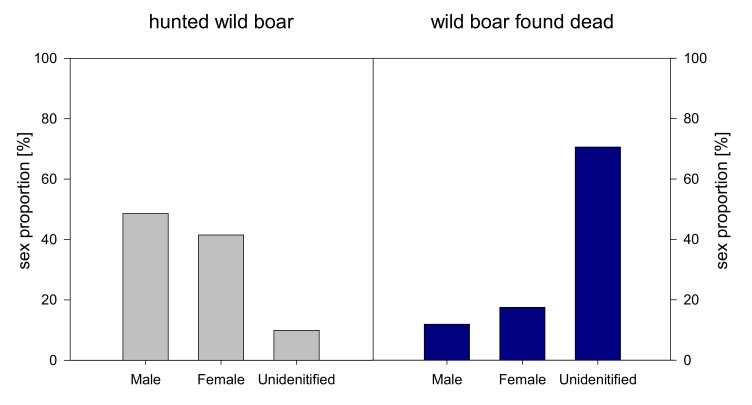
Sex distribution [%] of ASF positive wild boar hunted (grey bars, left side) and found dead (blue bars, right side) during 2014–2018 (cases of all years have been accumulated).

**Figure 4 vetsci-07-00015-f004:**
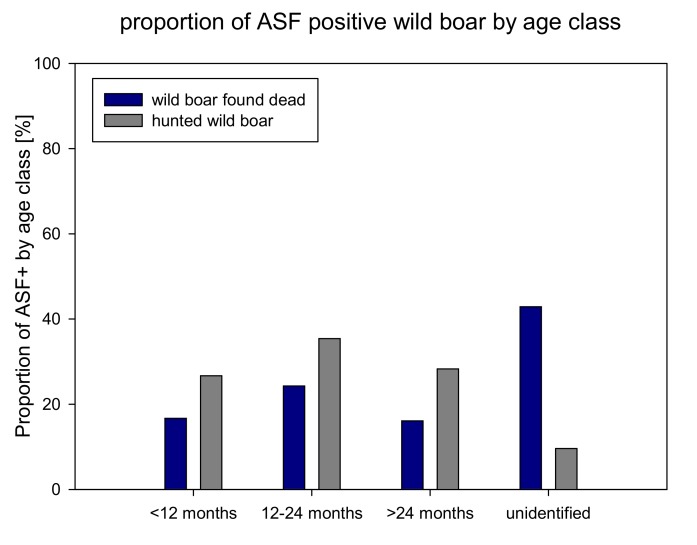
Proportion of ASF positive wild boar (hunted and found dead) by age class (2014–2018).

**Figure 5 vetsci-07-00015-f005:**
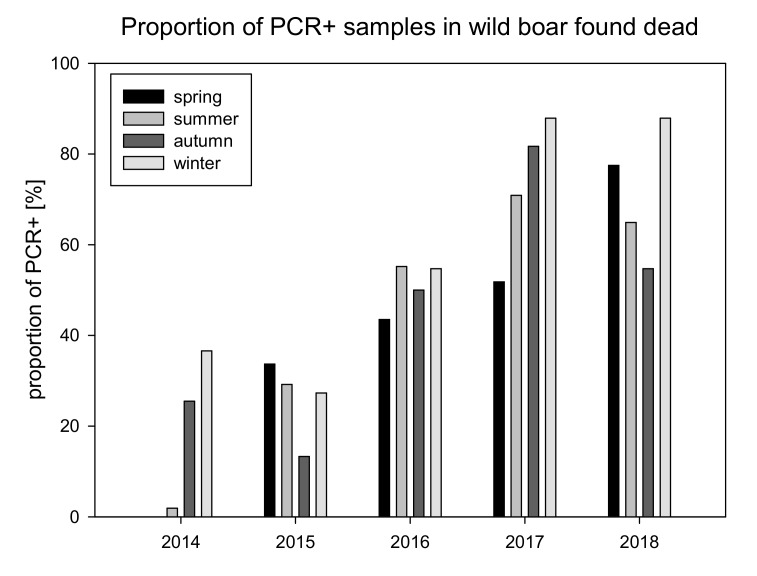
Proportion of PCR positive samples in found dead wild boar sorted by seasons (winter = December to February; spring = March to May; summer = June to August; autumn = September to November).

**Figure 6 vetsci-07-00015-f006:**
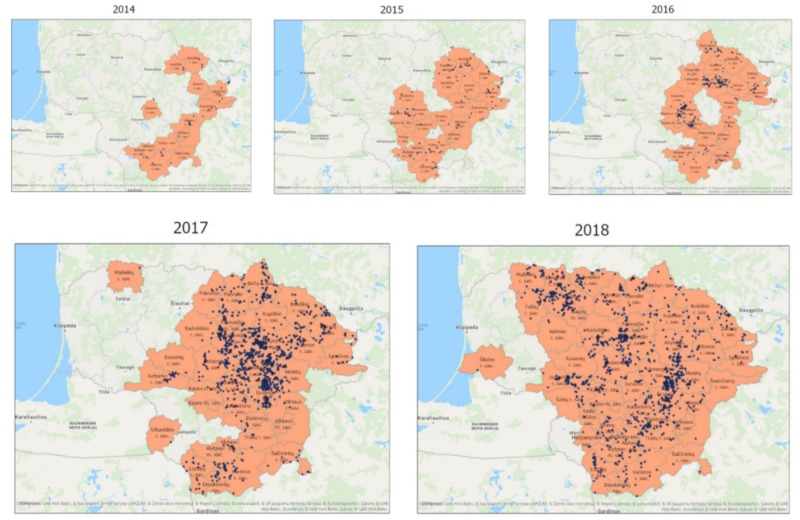
ASF spread in Lithuania (2014–2018). Affected municipalities and elderships are marked in orange. Positive cases in wild boar are indicated as blue dots. As soon as one case occurred within a year, the respective municipality was regarded as affected.

**Figure 7 vetsci-07-00015-f007:**
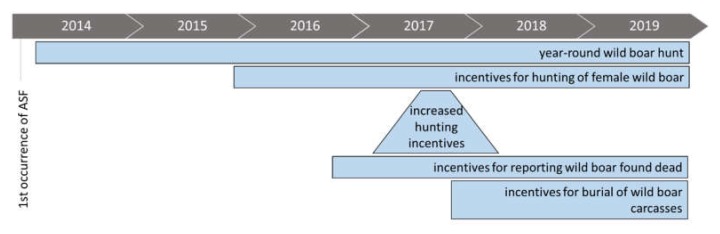
Timeline of disease control measures implemented in Lithuania (2014–2018).

**Table 1 vetsci-07-00015-t001:** African swine fever (ASF) outbreaks in domestic pigs and wild boar cases in Lithuania (2014–2018)**.**

Year	Outbreaks in Domestic Pigs	Wild Boar Cases	Affected Regional Municipalities	Territory Affected by ASF In Wild Boar *	
				Proportion of Affected Regional Municipalities [%]	Proportion Affected Elderships of Regional Municipalities [%]
2014	6	45	11	26	7
2015	13	110	18	40	14
2016	19	303	19	41	17
2017	30	1321	30	60	37
2018	51	1446	41	84	53
Total	119	3225	42	86	53

* Estimation based on area size of infected elderships and on area size of infected regional municipalities within restriction zones during the indicated year. Regional municipalities are the second level of administrative division and elderships are the smallest administrative division in Lithuania.

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
