# Peer review of "The African Swine Fever Epidemic in Wild Boar (*Sus scrofa*) in Lithuania (2014–2018)"

_vetsci, 2020, doi:10.3390/vetsci7010015_

Round 1

Reviewer 1 Report

The authors have provided satisfactory answers to the questions an the manuscript can be accepted

Author Response

The authors thank the reviewer for the time and helpful comments.

Reviewer 2 Report

The main concerns have been adressed by the uthors.

However, please change large-distance spread with long-distance spread.

In the conclusions they report that "during the first 3 years of infection the disease spread relatively slow..".

The Authors should explain compare to what or, if the comparison is with 2017, the sentence should be better expressed.

Author Response

(The authors gave the same response as above.)

Reviewer 3 Report

Dear authors,

In my opinion, the paper was well adapted and has been sufficiently improved for publication.

Some minor remarks:

Line 24: It has been ... is incorrect. The first occurrence is a stand-alone event, and present perfect tense should not be used. Replace by e.g. this was...

Line 32: ...territory are...: either territory is or territories are

Line 32: I would change to ASF cases in wild boar. Though clear, the sentence  as such is incorrect in English.

Line 58: ...nor assumed... should be nor did it assume

Line 65: remove 'an'

Line 273: ...according literature... should be according to literature

Though I leave this to the editors, some references could be improved. For example, Authority E.F.S. seems somewhat unlogical to me. EFSA or European Food Safety Authority would have my preference. Another example: in 23  <i>Sus scrofa</i>.

Author Response

The authors thank the reviewer for the time and helpful comments.

This manuscript is a resubmission of an earlier submission. The following is a list of the peer review reports and author responses from that submission.

Round 1

Reviewer 1 Report

Dear Authors,

Overall a well written paper discussing prevalence and spread of ASF in Lithuania. I have some remarks:

The title does not cover the load of the article. The abstract should be improved concerning English style. Line 30-32: I disagree, this is the only measure you discuss. Other measures could be discussed as well. First paragraph of the introduction should be improved concerning English style as well, the rest of the paper is overall well written. Line 41: 'Contained' seems to be a better word instead of 'controlled', if I understood the meaning correctly. Line 43-46: Where does this information come from? Either the source should be referenced or it should be moved to the results. Line 47-52: It would be interesting to discuss transmission routes. Material&Methods should be expanded. The experiment could not be repeated with this limited information. Line 83: Figure 6 should be Figure 1 and moved to the correct place in the text as it is mentioned as the first figure. 'Found dead wild boar' should be 'wild boar found dead'. The discussion should be improved, it contains a number of repetitions and results. The only measure discussed is the financial incentive, others could be discussed as well. You cannot claim it is the most effective if others are not researched. What other measures are or could be taken to prevent spread to wild boar? And to domestic pigs? Could you calculate an estimated prevalence among wild boar? And assess how much the population was thinned out? Line 181: I disagree with your statement. Active surveillance could be equally or more effective, but is harder for several reasons. Line 210: You did not research this, so cannot claim they were not identified. Line 211 and throughout the manuscript: an explanation about elderships is in order. By context it is clear what they are, but e.g. it is not clear what size compared to municipalities. Line 211-213: Is it not possible that in these elderships, ASF has just not been confirmed yet? It seems unlogical that some subpopulations have no contact with any others, while others do. Line 235-240: This statement is interesting, expand if possible. Line 242-245: It could be interesting to discuss possible reasons why. It could also be interesting to discuss how to avoid spread between wild boar, and to domestic pigs.

Author Response

Point 1: Overall a well written paper discussing prevalence and spread of ASF in Lithuania. I have some remarks:

The title does not cover the load of the article.

Response 1: The title has been checked.

Point 2: The abstract should be improved concerning English style.

Line 30-32: I disagree, this is the only measure you discuss.

Other measures could be discussed as well.

First paragraph of the introduction should be improved concerning English style as well, the rest of the paper is overall well written.

Response 2: We thank the reviewer for the helpful comments. Abstract and introduction have been revised accordingly and control measures are discussed in more detail in the discussion section.

Point 3: Line 41: 'Contained' seems to be a better word instead of 'controlled', if I understood the meaning correctly.

Line 43-46: Where does this information come from? Either the source should be referenced or it should be moved to the results.

Line 47-52: It would be interesting to discuss transmission routes.

Response 3: Adapted accordingly. Section removed.

Point 4: Material & Methods should be expanded. The experiment could not be repeated with this limited information. Line 83: Figure 6 should be Figure 1 and moved to the correct place in the text as it is mentioned as the first figure. 'Found dead wild boar' should be 'wild boar found dead'.

Response 4: Methodology of the calculation of spreading rate and the data set described. Figures are now mentioned in the correct order and headlines have been adapted accordingly.

Point 5: The discussion should be improved, it contains a number of repetitions and results.

The only measure discussed is the financial incentive, others could be discussed as well.

You cannot claim it is the most effective if others are not researched.

What other measures are or could be taken to prevent spread to wild boar? And to domestic pigs?

Could you calculate an estimated prevalence among wild boar? And assess how much the population was thinned out? Line 181: I disagree with your statement. Active surveillance could be equally or more effective, but is harder for several reasons.

Line 210: You did not research this, so cannot claim they were not identified.

Line 211 and throughout the manuscript: an explanation about elderships is in order. By context it is clear what they are, but e.g. it is not clear what size compared to municipalities. Line 211-213: Is it not possible that in these elderships, ASF has just not been confirmed yet? It seems unlogical that some subpopulations have no contact with any others, while others do.

Line 235-240: This statement is interesting, expand if possible.

Line 242-245: It could be interesting to discuss possible reasons why. It could also be interesting to discuss how to avoid spread between wild boar, and to domestic pigs.

Response 5: Line 181: Passive surveillance has been shown to be the most effective tool for ASF detection in wild boar. Several references are given in the text supporting this statement.

Line 210: has been removed.

Line 211: A short definition in the M&M section on what is an eldership and a municipality in Lithuania, including average/maximum/minimum size etc. described.

Line 211-213: The authors agree and added a clarification to the text.

Line 235-240: The section has been expanded accordingly.

Line 242-245: The possible reasons are discussed in the discussion section to keep the conclusions brief and concise.

Reviewer 2 Report

Mačiulskis et al. conducted the descriptive analysis for the African swine fever (ASF) outbreak occurred in Lithuania from 2014. The article tried to describe the ASF cases in wild boar more than ones in domestic pigs with information of test results in wild boar. Given the nature of the ASF outbreak in Europe, this information might be so helpful to understand the overview of the ASF outbreak in Lithuania. However, this manuscript should not be considered to be published in Veterinary Science as Article due to following reasons.

First of all, there are few academic findings. Most of the information does not provide scientific findings. It is not matching to the definition described at “Types of Publication”. The reviewer strongly recommends that the authors should perform the spatial and temporal analysis using the data in the manuscript. Unfortunately, there is detailed information (no methods, no figure/table) regarding the temporal analysis though the brief contents are in the manuscript (Line 138-141). Regarding the spatial analysis, the author should revise Figure 6 with emphasising annual spread of the ASF positive animals using the data of Supplementary table 3. The authors must consider to put necessary figures/tables or to put them adequate ways. In figure 1, the reason why shown in the figure is unclear (Table style is fine and is not space-wasting). Figure 2 is not satisfying the scientific style. The author can change the style of Figure 3 and 4 to the cumulative bar chart. Descriptions of figure legends in most Figures are poor. Several statements in Discussion are confusing. Contents in Line 180-184 is not adequate for the discussion regarding the data in 2016 because the outbreak had already moved to the middle phase. The authors concluded that the disease had spread more quickly from 2017 in Line 242-245 though the author had suggested that the increase of reporting in 2017 should be due to willingness of reporting by the incentives implemented. Throughout the manuscript except for Discussion and Conclusion, there are many sentences too hard to be understood in respect of English grammar.

Author Response

Point 1: Mačiulskis et al. conducted the descriptive analysis for the African swine fever (ASF) outbreak occurred in Lithuania from 2014. The article tried to describe the ASF cases in wild boar more than ones in domestic pigs with information of test results in wild boar. Given the nature of the ASF outbreak in Europe, this information might be so helpful to understand the overview of the ASF outbreak in Lithuania. However, this manuscript should not be considered to be published in Veterinary Science as Article due to following reasons.

Response 1: As mentioned in the title, the manuscript focused on ASF cases in Lithuanian wild boar. Outbreaks in domestic pigs are of great interest and need further investigation but this was not the scope of the presented study.

Point 2: First of all, there are few academic findings. Most of the information does not provide scientific findings. It is not matching to the definition described at “Types of Publication”. The reviewer strongly recommends that the authors should perform the spatial and temporal analysis using the data in the manuscript.

Response 2: Spatial and temporal analysis have been performed as described in the M&M section.

Point 3: Unfortunately, there is detailed information (no methods, no figure/table) regarding the temporal analysis though the brief contents are in the manuscript (Line 138-141). Regarding the spatial analysis, the author should revise Figure 6 with emphasising annual spread of the ASF positive animals using the data of Supplementary table 3.

Response 3: Temporal analysis is described in the M&M section.

Point 4: The authors must consider to put necessary figures/tables or to put them adequate ways. In figure 1, the reason why shown in the figure is unclear (Table style is fine and is not space-wasting). Figure 2 is not satisfying the scientific style. The author can change the style of Figure 3 and 4 to the cumulative bar chart. Descriptions of figure legends in most Figures are poor.

Response 4: Figure legends have been improved accordingly.

Point 5: Several statements in Discussion are confusing. Contents in Line 180-184 is not adequate for the discussion regarding the data in 2016 because the outbreak had already moved to the middle phase. The authors concluded that the disease had spread more quickly from 2017 in Line 242-245 though the author had suggested that the increase of reporting in 2017 should be due to willingness of reporting by the incentives implemented.

Response 5: The discussion has been adapted.

Point 6: Throughout the manuscript except for Discussion and Conclusion, there are many sentences too hard to be understood in respect of English grammar.

Response 6: English grammar has been improved.

Reviewer 3 Report

This paper is a very interesting and informative report on the ASF epidemic in Lithuanian wild boars from 2014 to 2018.

The paper is very well written, very clear and reports interesting descriptive findings for the scientific community working on ASF.

I have no major remark on the paper. My only recommendations would be:

to add in the material and methods section the description of the methodology used to estimate the speed of ASF spread at the country level. There is no detail on this. describe also the biosecurity measures that were implemented by hunters or people searching wild boar carcasses

Author Response

Dear Reviewer please find attache file with Author's reply to your comments.

Reviewer 4 Report

scientific terminoligy shall be improved: examples: spreding pattern with "huge jump"; "fulminant" epidemic: what do they mean?

in the abstract is stated that the study contributes to a better understanding of the disease but after reading the discussion and conclusion the achievment, in terms of improved knowledge, is not clear

the sentence reported in line 53 - 54 should be supported by a reference.

The use of "however" on line 170 should be revised.

Conclusions are not supported by the data reported in the manuscript, and in general, an analysis of the data is missing. Example 1: why they consider that the disease has shown "low contagiosity": this is also in contraxt with the fact that they are reporting that in Lithuania the disease is spreading fastern than in other countries.. example 2: they are reporting no seasonality but this is not supported by an analysis..

The data should be analised in a more sistematic manner and the discussion and conclusion in line with the data reported

Author Response

Dear Reviewer, please find author's reply to your comments.

Reviewer 5 Report

It is better to indicate the domestic outbreaks and wild boar cases for each year, togheter with cumulated cases.

It is also necessary to report the denominators, both domestic and wild boar.

It is important to describe the distribution of these denominators, at least for each administrative division.

It is useful, for each year, to describe your analysis distinguishing the infected area from the non infected area.

It is useful to describe, even in a general way, the trend of the hunting activity for each month, even just with a graph.

Author Response

Dear reviewer, please find author's reply to your comments.
